# Potential gains in life expectancy from reducing amenable mortality among people diagnosed with serious mental illness in the United Kingdom

Alex Dregan [1]*, Ann McNeill[1], Fiona Gaughran[1,2], Peter B. Jones [3,4], Anna Bazley[2], Sean Cross[2], Kate Lillywhite[2], David Armstrong[5], Shubulade Smith[1], David P. J. Osborn[6], Robert Stewart[1,2], Til Wykes[1], Matthew Hotopf[1,2]

1 Institute of Psychiatry, Psychology and Neuroscience (IoPPN), King's College London, London, United Kingdom, 2 South London and Maudsley NHS Foundation Trust and London, London, United Kingdom, 3 Department of Psychiatry, University of Cambridge, Cambridge, United Kingdom, 4 Cambridgeshire and Peterborough NHS Foundation Trust, Cambridge, United Kingdom, 5 School of Population Health and Environmental Sciences, King's College London, London, United Kingdom, 6 Division of Psychiatry, Faculty of Brain Sciences, University College London, London, United Kingdom

* alexandru.dregan@kcl.ac.uk

## Abstract

### Background

To estimate the potential gain in life expectancy from addressing modifiable risk factors for all-cause mortality (excluding suicide and deaths from accidents or violence) across specific serious mental illness (SMI) subgroups, namely schizophrenia, schizoaffective disorders, and bipolar disorders in a Western population.

### Methods

We have used relative risks from recent meta-analyses to estimate the population attribution fraction (PAF) due to specific modifiable risk factors known to be associated with all-cause mortality within SMI. The potential gain in life expectancy at birth, age 50 and age 65 years were assessed by estimating the combined effect of modifiable risk factors from different contextual levels (behavioural, healthcare, social) and accounting for the effectiveness of existing interventions tackling these factors. Projections for annual gain in life expectancy at birth during a two-decade was estimated using the Annual Percentage Change (APC) formula. The predicted estimates were based on mortality rates for year 2014–2015.

### Results

Based on the effectiveness of existing interventions targeting these modifiable risk factors, we estimated potential gain in life expectancy at birth of four (bipolar disorders), six (schizoaffective disorders), or seven years (schizophrenia). The gain in life expectancy at age 50 years was three (bipolar disorders) or five (schizophrenia and schizoaffective disorders)

**Data Availability Statement:** All relevant data are within the manuscript and its Supporting Information files.

**Funding:** Alex Dregan work is supported by the Medical Research Council (grant number MR/S028188/1). This paper represents independent research part-funded by the National Institute for Health Research (NIHR) Biomedical Research Centre at South London and Maudsley NHS Foundation Trust and King's College London. Fiona Gaughran is in part supported by the National Institute for Health Research's (NIHR) Biomedical Research Centre at South London and Maudsley NHS Foundation Trust and King's College London, the Stanley Medical Research Institute, the Maudsley Charity and the National Institute for Health Research (NIHR) Applied Research Collaboration South London (NIHR ARC South London) at King's College Hospital NHS Foundation Trust. Til Wykes acknowledges the support of the NIHR BRC Research Centre at and her NIHR Senior Investigator award. The views expressed are those of the author(s) and not necessarily those of the NIHR or the Department of Health and Social Care.

**Competing interests:** Matthew Hotopf is principal investigator of the RADAR-CNS programme, a precompetitive public private partnership funded by the Innovative Medicines Initiative and European Federation of Pharmaceutical Industries and Associations (EFPIA). The programme receives support from Janssen, Biogen, MSD, UCB and Lundbeck. In the last 3 years, Fiona Gaughran has received support or honoraria from Lundbeck, Hikma, Otsuka and Sunovion, and has a family member who has professional links to Lilly and GSK, including shares.

years. The projected gain in life expectancy at age 65 years was three (bipolar disorders) or four (schizophrenia and schizoaffective disorders) years.

## Conclusions

The implementation of existing interventions targeting modifiable risk factors could narrow the current mortality gap between the general and the SMI populations by 24% (men) to 28% (women). These projections represent ideal circumstances and without the limitation of overestimation which often comes with PAFs.

## Introduction

While life expectancy of the general population has increased steadily over the past century, people with serious mental illness (SMI), including diagnoses of schizophrenia, schizoaffective disorder, or bipolar disorder, have a reduced life expectancy of 13 years for men and 12 years for women.[1] Around a fifth (20%) of the excess mortality in people with SMI is due to non-natural deaths such as suicide and accidents but most deaths can be ascribed to common diseases such as cardiovascular disease (CVD), respiratory illnesses, type 2 diabetes (T2DM), cancer, and digestive disorders.[2–4] These diseases, in their turn, are partly attributable to modifiable unhealthy lifestyle factors (e.g. smoking, sedentarism, obesity, dyslipidemia), adverse social context (e.g. social isolation, social deprivation), and suboptimal healthcare use and efficiency (e.g. prevention, treatment adherence).[5–11] These modifiable risk factors therefore provide opportunities for interventions to reduce the excess premature mortality experienced by people with SMI.[7, 12, 13] Currently, there is limited evidence about the potential gain in life expectancy among people with SMI by addressing multiple modifiable risk factors for all-cause mortality. In this study we have quantified the potential gain in life expectancy at birth and adult (ages 50 and 65) years that might realistically be achieved by evaluating the population attributional fraction due to specific modifiable risk factors and adjusting these estimates by the published effectiveness of interventions from international studies to address these factors. Projection of the expected annual increase in life expectancy at birth over future decades can help monitor progress towards meeting set goals across specific SMI groups. Therefore, this paper employed the life table method to estimate potentially attainable gain in life expectancy over the life course for specific SMI subgroups and projected the gains over a two-decade period.

## Materials and methods

### Data source

The existing literature was searched to estimate the prevalence of modifiable risk factors for natural causes of all-cause mortality within the SMI population, the relative risk (RR) for all-cause mortality attributed to those risk factors, and the effectiveness of health interventions targeting modifiable risk factors within the SMI population. SMI was broadly defined to include schizophrenia, schizoaffective disorders, and bipolar disorders. We searched PubMed and reviews-focused electronic databases (e.g. Cochrane Database of Systematic Reviews, DARE, DoPHER) to identify the most recent meta-analyses with relevant quantitative data between 2008 and 2018. Where no meta-analyses were identified, data from the most recent observational studies with representative populations were used. To facilitate comparison,

priority was given to meta-analyses and observational studies with comparative data available across all three SMI subgroups. Where schizophrenia and schizoaffective disorders were grouped together or when only overall estimates were provided (rather than SMI subgroup specific estimates), we have assumed the same estimate within each SMI subgroup. A description of the resources used to derive the study estimates is provided in S1 Table in S1 File.

**Serious mental illness.** There is inconsistent definition of SMI in practice, [14–16] however, the term commonly refers to people diagnosed with schizophrenia, schizoaffective disorders, and bipolar disorder.[1, 2, 17] Major depressive disorders are also considered SMI, but not consistently, and here we have decided to use a narrower definition and consider SMIs with superior evidence about drivers of excess mortality.[18] While these disorders share some common characteristics (i.e. overlap in clinical and genetic attributes), core risk factors and associations, they also possess distinctive clinical features and healthcare burden.[19–24] For instance, depression and panic disorders were suggested to be more severe within bipolar disorder, while lack of insight and early age onset were more common within schizophrenia.[20, 23] Since these distinctive clinical features may lead to differences in comorbidity rates, the three SMI disorders may also present with distinct mortality associations. We therefore considered schizophrenia, schizoaffective disorders, and bipolar disorders as separate entities for the present study, though we acknowledge the substantial overlap in the probable drivers of excess mortality.[25]

**Modifiable risk factors.** The present study focused on natural causes of death and was restricted to risk factors known to be amenable and preventable within the SMI population. In the context of the present study, a modifiable risk factor was defined as an event known to directly or indirectly increase the probability of death and for which there are proven methods to reduce the impact on all-cause mortality in people with SMI.[26] These factors include behavioural risk factors (e.g. smoking, physical inactivity, obesity, poor diet, substance abuse) and healthcare-related factors (e.g. uptake of public health interventions, uptake of effective treatment, access to healthcare resources), as well as upstream socioeconomic determinants (e.g. social exclusion, stigma, deprivation). A multicontextual view of avoidable mortality was favoured since it spans the different levels of explanation for the reduced life expectancy within the SMI population as defined here. It also facilitates the identification of proven interventions to reduce the mortality gap between people with and without SMI.[27]

## Statistical analysis

Prevalence of modifiable risk factors, relative risk for all-cause mortality, and the effectiveness of existing interventions were identified through systematic searches of published data. To determine the proportion of mortality burden attributable to each modifiable risk factor we computed the population-attributable fraction (PAF) from the prevalence ($P_{cs}$) of modifiable risk factors and unadjusted relative risk ($RR_{cs}$) of all-cause mortality associated with each risk factor separately, using the formula:

$$PAF = P_{cs}(RR_{cs}-1)/P_{cs}(RR_{cs}-1) + 1 \qquad (1)$$

The PAFs for all-cause mortality associated with a modifiable risk factor were computed using the prevalence and RR estimates obtained from meta-analysis or individual studies. The PAF is interpreted as the proportion of mortality risk that could be eliminated from the SMI population, if the exposure to the risk factor was eliminated or reduced. To allow for the overlap between different modifiable risk factors in influencing all-cause mortality, we have also estimated the combined impact of multiple modifiable risk factors within each group of

determinants (e.g. behavioural, healthcare, social) and across all modifiable risk factors using, [28]

$$\text{PAF}_{combined} = 1\text{-}\pi(1\text{-}\text{PAF}_r) \qquad (2)$$

where r is the risk factor index. While the formula assumes no interaction effects, it ensures that the PAF for the combined contribution of modifiable risk factors to all-cause mortality is not greater than 1.[28]

To estimate likely gain in life expectancy from specific interventions in SMI, we adjusted the PAF value of modifiable risk factors by the effectiveness of interventions aimed at reducing the rate of modifiable risk factors within the SMI population. For example, if the PAF for smoking was 46% (based on (1) above), then this figure was weighted by the effectiveness of smoking cessation interventions in people with SMI. If the effectiveness of smoking cessation among people with SMI would be around 36%,[29] then the likely gain in life expectancy at birth with respect to smoking would be based on the assumption that 17% (0.46 weighted by 0.36) of smoking-related mortality risk could be potentially eliminated from people with schizophrenia. Data on the effectiveness of interventions were only available at aggregate level.

Abridged period life table, based on the Chiang's method,[30] for 2014–2015 were used to describe the base case population for life expectancy and all-cause mortality rates in United Kingdom (UK) adults with SMI. The 2014–2105 period represents the most recent data (2018) released by the National Health Service Outcome Framework Indicators (NHS QoF),[31] which includes aggregated mortality rate and population data into 5-year age intervals for SMI adults 18 to 74 years of age. For SMI adults aged 75 years or older, the mortality rates were calculated based on 2014–2015 mortality data among adults of similar age from the general population. Specifically, we have multiplied the mortality rate for 2014–2015 among adults aged 75 years or over from the general population by the increased mortality risk among adults aged 75 years or over with SMI (e.g. 2.5 for 75–79, 1.9 for 80 to 84, and 1.3 for 85+ year-group) using published data.[32] For instance, the mortality rate in 2014–2105 for SMI adults aged 75 to 79 years was estimated to be 2.5 times greater (0.0835) than the 0.0334 mortality rate experienced by 75–79 years old adults from the general population during 2014–2015. Because younger people (<15 years of age) are unlikely to receive a diagnosis of SMI,[1] we have used the UK all-cause mortality rates for the under 18 year age group in substitution. Separate abridged life tables were constructed for each modifiable risk factor. These mortality rates were used to create a cause-eliminating life table to estimate the potential gain in life expectancy at birth if modifiable risk factors could be reduced or eliminated within specific SMI subgroups.[33] The gain in life expectancy represents the change in life expectancy at birth that would be obtained under the hypothetical reduction of the cause (modifiable risk factors) of premature mortality within specific SMIs subgroups. Let 65.9 years denote the life expectancy at birth (age 0) from actual life table for a person with schizophrenia, and 68.3 year denote life expectancy at birth from smoking(cause)-eliminated life table for the same person. Then the gain in life expectancy at birth due to smoking(cause) elimination or reduction is 2.4 years (68.3–65.9). This approach has been widely employed to estimate the potential gain in life expectancy if a specific cause could be reduced or eliminated.[34] To capture potential gain in life expectancy across the life course and to allow for the fact that people are diagnosed with SMI at different life stages,[35] we have estimated two additional scenarios: (I) potential gain in life expectancy among people with SMI who survive to age 50, and (II) potential gain in life expectancy among people with SMI who survive to age 65. For (I), the analyses estimated the average remaining years of life that SMI survivors to age 50 were expected to live if mortality levels at

each age-group over 50 remained constant.[36] A similar approach was used for age 65 (II). [37]

To quantify the expected average annual amount of change in life expectancy, we have divided the total gain in life expectancy at birth by the number of years during a specific period. Given the stable mortality rates and modest change in life expectancy over the past decade within the general population,[38] we have considered a 20-year period as a reasonable timeframe for projecting likely average annual amount of change in life expectancy among people with SMI. In addition, we computed the arithmetic average percentage of change in life expectancy during a 20-year period by dividing the average amount of annual change by the expectancy value at the beginning of the period, using the following formula: [33]

1. $L_{ex2030} / L_{ex2018} = e^{rt(20)}$ (3)

2. transform (1) in ln: $\ln(L_{ex2038} / L_{ex2018}) = rt$

3. dividing (2) by (1): $\ln(L_{ex2038} / L_{ex2018})/t = r$

Where, $L_{ex2038}$, represent life expectancy in 2038, and $L_{ex2018}$ refers to life expectancy in 2018., r represents the Annual Percentage Change (APC), and t the number of years. In sensitivity analyses, we applied average all-cause mortality rates over a five-year period (2009/2010 to 2014/2015) using mid-interval (2011/2012) population as the denominator to assess the robustness of our findings. We used Stata vers. 15 and Excel to analyse the data.

## Results

Table 1 shows the estimates for the prevalence of modifiable risk factors within specific SMI subgroups. According to most recent meta-analyses,[39, 40] around 59% of people with schizophrenia and 49% of those with bipolar disorder are current smokers. Accelerometry-based evidence indicated that sedentary behaviour was common among 81% of people with schizophrenia and 78% among those with bipolar disorders[41, 42]. The uptake of public health interventions (i.e. screening, preventative initiatives) was also poor, with around 50% of people with SMI not being screened for CVD risk factors.[39] Low rates of receiving specialist treatment following a major life event were identified, with 47% of the SMI population not receiving revascularisation procedures following a myocardial infarction, for instance.[43] Regarding social context, around 65% and 55% of people with schizophrenia or bipolar disorders, respectively, reported the experience of stigma.[44, 45]

### Lifestyle risk factors

Table 2 shows potential gains in life expectancy at birth, and ages 50 and 65 years associated with tackling modifiable risk factors for all-cause mortality in specific SMIs. Smoking-related conditions accounted for 46% of all deaths in schizophrenia and 22% of all deaths in bipolar disorders (based on formula (1) above). Based on age-aggregated effectiveness of existing smoking-cessation interventions, smoking-targeted initiatives have the potential to extend the life expectancy at birth by on average two years and five months within schizophrenia or schizoaffective disorders, and by one year and one month within bipolar disorders. Potential gain in life expectancy at age 50 related to smoking-targeted interventions, was two years within schizophrenia or schizoaffective disorders, and nine months among bipolar disorder. Likewise, potential gain in life expectancy at age 65 from smoking-targeted interventions was one year and seven months within schizophrenia or schizoaffective disorders, and seven months among bipolar disorder. Sedentary behaviour was the most potent determinant of avoidable mortality within bipolar disorders, accounting for just over a third (34%) all-cause mortality (using

**Table 1. Prevalence of modifiable risk factors within specific SMIs and the general population.**

|  | Schizophrenia | | Bipolar disorder | | Schizoaffective disorders | |
|---|---|---|---|---|---|---|
|  | % | RR | % | RR | % | RR |
| **LIFESTYLE FACTORS** | | | | | | |
| Smoking | 59 | 2.45 | 49 | 1.57 | 59 | 2.50 |
| Sedentary | 81 | 1.66 | 78 | 1.66 | 81 | 1.66 |
| Diet | 69 | 1.53 | 69 | 1.53 | 69 | 1.53 |
| Obesity | 50 | 1.47 | 50 | 1.47 | 50 | 1.47 |
| Substance abuse | 47 | 1.78 | 45 | 1.89 | 54 | 1.78 |
| Metabolic syndrome | 33 | 2.10 | 32 | 1.70 | 35 | 2.10 |
| **HEALTHCARE FACTORS** | | | | | | |
| Clozapine/Lithium | 24 | 1.88 | 29 | 1.70 | 24 | 1.88 |
| Healthcare access | 46 | 1.34 | 46 | 1.11 | 46 | 1.34 |
| Treatment disparity (e.g. revascularisation) | 47 | 1.15 | 47 | 1.15 | 47 | 1.15 |
| Screening uptake | 50 | 1.18 | 50 | 1.18 | 50 | 1.18 |
| **SOCIAL FACTORS** | | | | | | |
| Social deprivation | 30 | 1.90 | 26 | 1.30 | 26 | 1.90 |
| Social isolation | 32 | 1.19 | 21 | 1.19 | 32 | 1.19 |
| Stigma experience | 65 | 1.12 | 55 | 1.12 | 49 | 1.12 |

RR–relative risk of mortality given the risk factor

formula (1) above). Physical inactivity-focused interventions could extend life expectancy at birth among the SMI subgroups by on average one year and three months. Under an additive scenario, tackling behavioural and metabolic determinants of avoidable mortality would lead to a potential gain in life expectancy at birth of four (bipolar disorders) or six (schizophrenia) years. The combined gain in life expectancy at age 50 years was three (bipolar disorders) or five (schizophrenia) years, while at age 65 years it was two (bipolar disorders) or four (schizophrenia) years.

## Healthcare systems determinants

Table 2 illustrates that ineffective access to healthcare resources accounted for a larger proportion of all-cause mortality within schizophrenia and schizoaffective disorders (14%) relative to bipolar disorders (5%). Allowing for the effectiveness of current interventions, the potential gain in life expectancy at birth by tackling inadequate healthcare access was three (bipolar disorders) or seven (schizophrenia and schizoaffective disorders) months. The figures in Table 2 have been weighted to reflect the fact that only 30% of people with schizophrenia (treatment-resistant patients) are currently prescribed clozapine. Consequently, improved clozapine prescribing among eligible patients with schizophrenia and improved lithium prescribing within bipolar disorder patients could lead to potential gains in life expectancy at birth of around three months or one year, respectively. Additional gain in life expectancy at age 50 associated with clozapine was two months (schizophrenia) and eight months for lithium (bipolar disorders). Under an additive scenario, improving healthcare practices and processes could increase life expectancy at birth by seven or eight months within the SMI population.

## Social determinants

Within social determinants, social deprivation accounted for the largest proportion of avoidable mortality within schizophrenia (21%), schizoaffective disorders (19%), and bipolar

**Table 2. Potential gains in life expectancy at birth and ages 50 and 65 years associated with modifiable risk factors for all-cause mortality in specific SMIs.**

| | | Schizophrenia | | | | Bipolar disorders | | | | Schizoaffective disorders | | | |
|---|---|---|---|---|---|---|---|---|---|---|---|---|---|
| | ES (%) | PAF (%) | LYG–age Birth | 50 | 65 | PAF (%) | LYG- age Birth | 50 | 65 | PAF (%) | LYG- age Birth | 50 | 65 |
| **LIFESTYLE MODEL** | | | | | | | | | | | | | |
| Smoking | 36 | 46 | 2.5 | 2.1 | 1.7 | 22 | 1.1 | 0.9 | 0.7 | 46 | 2.5 | 2.1 | 1.7 |
| Sedentary | 25 | 35 | 1.3 | 1.0 | 0.8 | 34 | 1.3 | 1.0 | 0.8 | 35 | 1.3 | 1.0 | 0.8 |
| Diet | 31 | 27 | 1.1 | 0.9 | 0.7 | 27 | 1.1 | 0.9 | 0.7 | 27 | 1.1 | 0.9 | 0.7 |
| Obesity | 24 | 19 | 0.7 | 0.6 | 0.5 | 19 | 0.7 | 0.6 | 0.5 | 19 | 0.7 | 0.6 | 0.5 |
| Substance abuse | 17 | 27 | 0.7 | 0.6 | 0.5 | 29 | 0.7 | 0.6 | 0.5 | 30 | 0.7 | 0.6 | 0.5 |
| **COMBINED[b]** | | **85** | **5.4** | **4.4** | **3.6** | **78** | **3.8** | **3.1** | **2.5** | **85** | **5.4** | **4.4** | **3.6** |
| Metabolic syndrome | 30 | 27 | 1.1 | 0.9 | 0.7 | 18 | 0.8 | 0.6 | 0.4 | 28 | 1.1 | 0.9 | 0.7 |
| **COMBINED (lifestyle)** | | **89** | **5.8** | **4.7** | **3.8** | **82** | **3.8** | **3.0** | **2.4** | **89** | **5.8** | **4.7** | **3.8** |
| **HEALTHCARE MODEL** | | | | | | | | | | | | | |
| Clozapine/Lithium[b] | 13/38 | 17 | 0.3 | 0.2 | 0.2 | 17 | 0.9 | 0.7 | 0.6 | 17 | 0.3 | 0.2 | 0.2 |
| Healthcare access | 38 | 14 | 0.7 | 0.6 | 0.5 | 5 | 0.3 | 0.2 | 0.2 | 14 | 0.7 | 0.6 | 0.5 |
| Treatment disparity | 37 | 7 | 0.4 | 0.3 | 0.3 | 7 | 0.4 | 0.3 | 0.3 | 7 | 0.4 | 0.3 | 0.3 |
| Screening uptake | 41 | 8 | 0.4 | 0.3 | 0.3 | 8 | 0.4 | 0.3 | 0.3 | 8 | 0.4 | 0.3 | 0.3 |
| **COMBINED** | | **37** | **0.7** | **0.5** | **0.5** | **36** | **0.8** | **0.6** | **0·5** | **37** | **0.7** | **0.5** | **0.5** |
| **SOCIAL MODEL** | | | | | | | | | | | | | |
| Social deprivation | 23 | 21 | 0.7 | 0.6 | 0.5 | 7 | 0.3 | 0.2 | 0.2 | 19 | 0.5 | 0.4 | 0.4 |
| Social exclusion | 38 | 6 | 0.3 | 0.2 | 0.2 | 4 | 0.3 | 0.2 | 0.2 | 6 | 0.3 | 0.2 | 0.2 |
| Stigma experience | 24 | 7 | 0.3 | 0.2 | 0.2 | 6 | 0.1 | 0.1 | 0.1 | 6 | 0.1 | 0.1 | 0.1 |
| **COMBINED** | | **30** | **0.4** | **0.3** | **0.3** | **16** | **0.1** | **0.1** | **0.1** | **28** | **0.2** | **0.2** | **0.2** |
| ***ATTAINABLE[c]*** | | **95** | **6.6** | **5.2** | **4.4** | **90** | **4.2** | **3.3** | **2.7** | **95** | **6.4** | **5.1** | **4.3** |
| ***APC[d]*** | | **5%** | **0.3** | **0.3** | **0.2** | **5%** | **0.2** | **0.2** | **0.1** | **5%** | **0.3** | **0.3** | **0.2** |

ES- effectiveness of existing interventions at reducing the rate of risk factors. PAF–population attributable fraction; LYG–life years gained at specific ages from reducing a cause weighted by the ES

[a] Combined = comorbidity adjusted gain in life expectancy for specific determinants

[b] Lithium is prescribed for bipolar disorder, mainly. Clozapine has been adjusted to reflect that only a third of patients are eligible for prescribing

[c] Attainable = potential gain in life expectancy considering the combined effect of multiple risk factors

[d] APC–annual percentage change, together with amount of annual change (rounded data) in life expectancy.

disorders (7%). Tackling social deprivation-related inequalities within the SMI population has the potential to increase the life expectancy at birth by on average three (bipolar disorders), five (schizoaffective disorders), or seven (schizophrenia) months. Under an additive scenario, tackling social determinants of all-cause mortality within SMI may increase life expectancy at birth on average by one (bipolar disorders), two (schizoaffective disorders), or four (schizophrenia) months. The potential gain in life expectancy at ages 50 and 65 years from tackling social deprivation was one (bipolar) to three (schizophrenia) months.

When estimated collectively, behavioural habits, healthcare practices, and social determinants accounted for 90% (bipolar disorders) and 95% (schizophrenia and schizoaffective disorders) of all-cause mortality. Allowing for the effectiveness of existing interventions, tackling these modifiable risk factors would lead to a potential gain in life expectancy at birth of four years within bipolar disorders, six years within schizoaffective disorders, or seven years within schizophrenia. Assuming similar APC in life expectancy within men and women with schizophrenia, the estimated life expectancy gain would gradually narrow the current life expectancy gap on average by three years for men and four years among women during a 20-year period (Fig 1). Sensitivity analyses using the average five-year mortality rate produced only marginally

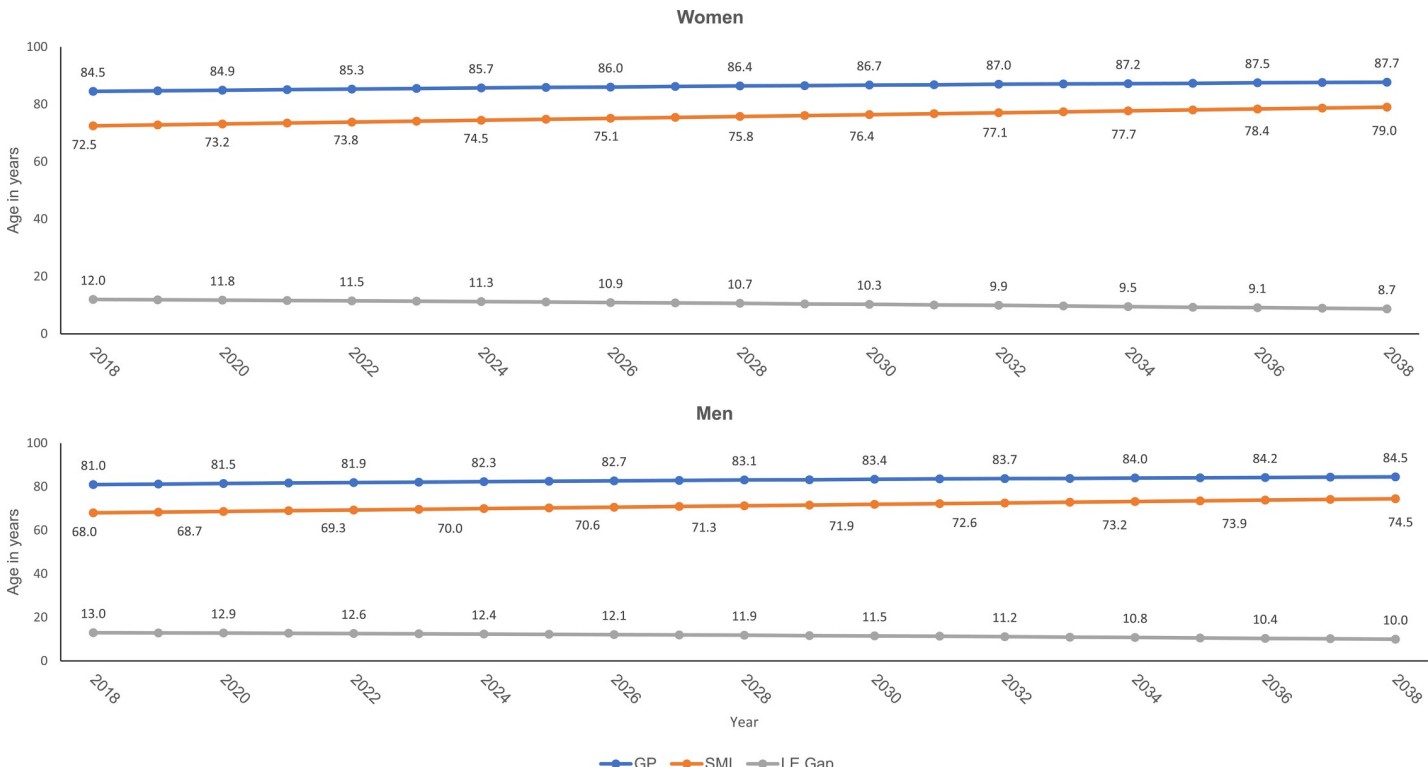

**Fig 1. Trends in life expectancy gain at birth and projected gap (LE Gap) between the general population (GP) and the schizophrenia (SMI) population over a two-decade period.**

lower estimates for gain in life expectancy across the life course validating the robustness of our findings (S2 Table in S1 File).

## Discussion

The present study employed a multicontextual approach to quantify potential gains in life expectancy from tackling modifiable causes of premature mortality within specific SMI sub-groups. The present study modeling approach relied on two assumptions that (a) the estimated average effectiveness of interventions represented all studies and countries, and (b) that mortality rates of UK adults with SMI applied universally. Accounting for the combined effects of multiple modifiable risk factors, our findings indicated that an attainable target for gain in life expectancy at birth among people with SMI would be around six and a half years for schizophrenia and schizoaffective disorders, and around four years within bipolar disorders. When projected during a 20-year period, these estimates translated into an average annual percentage change in life expectancy at birth of between two to three months (5%). Assuming similar gain in life expectancy at birth among men and women with SMI and using national forecasts for the UK general population, we projected 24% (men) to 28% (women) narrowing of the mortality gap between the general and the SMI populations over a 20-year target. We also estimated that tackling modifiable risk factors for natural causes of death could extend life expectancy at age 50 years three (bipolar disorders) or five (schizophrenia or schizoaffective disorders) years. In addition, tackling modifiable risk factors could extend life expectancy at age 65 years by three (bipolar disorders) or four (schizophrenia and schizoaffective disorders) years.

The implementation of lifestyle interventions depends also on achieving increased acceptance and participation rates on the part of patients. SMIs are commonly associated with impairment in patients' awareness about the implications of unhealthy behaviours or treatment non-adherence to their health and wellbeing. Poor insight into the illness and its treatment, may challenge clinicians' capacity to convince SMI patients to adopt healthy behaviours or adhere to treatment. These suggestions are supported by recent evidence documenting that people with SMI failed to manifest similar reduction in mortality rates due to natural causes observed in the general populations over the past decades.[46] Future studies are needed to identify how best to enhance SMI patients' awareness about the benefits of treatment compliance and lifestyle changes to their health and quality of life.

Smoking emerged as the best single modifiable candidate for increasing life expectancy within schizophrenia (2.4 years), while sedentary behaviour (1.2 years) appeared to be the best single modifiable candidate for increasing life expectancy within bipolar disorder. Yet, lifestyle behaviours within the SMI population continue to be marginalised and poorly integrated into care pathways.[47] For instance, health care professionals were often reluctant to engage in smoking cessation behaviours[48] and smoking cessation services may be less accessible to people with SMI.[49] On the positive side, however, people with SMI appeared equally motivated to want to change negative lifestyle behaviours [50]–though interventions may be less effective.[51] A recent trial illustrated, for instance, that a bespoke smoking cessation intervention embedded in routine mental health care settings[52] was associated with a 56% greater reduction in smoking rates compared to usual care within people with schizophrenia and bipolar disorder. Other studies suggested, however, lower rates of smoking cessation.[53] Regardless of these variation, such findings lend support to our proposed gain in life expectancy within the SMI population, if the effectiveness of current lifestyle interventions can be maintained or improved in the long-term.

Despite valid concerns[54–58] about a potential association between antipsychotic drugs with increased risk of metabolic disorders, previous studies revealed a potential for clozapine and lithium to reduce premature mortality among people with schizophrenia and bipolar disorders.[59–61] This may seem counter-intuitive given that clozapine is prone to cause obesity, diabetes, cardiomyopathy[58] and, rarely, agranulocytosis, as well as being reserved for the most severe, treatment resistant group. Also, lithium has a narrow therapeutic range, and is potentially associated with renal impairment or failure.[62,63] Prior research suggest that some of these side-effects may be offset by the evidence for reduced all-cause mortality. For instance, studies have shown up to 50% reduction in all-cause mortality among people with schizophrenia receiving clozapine compared to those never prescribed the drug.[64] The mechanisms through which these drugs have their beneficial effects may be a direct consequence of treatment on mental health symptoms, or on service related factors such as continued monitoring and continuity of care which is necessitated by the routine monitoring involved. While effective, clozapine and lithium are not always accepted by patients and require expertise and experience in order to be prescribed safely. Thus, there are implementation challenges to any policy highlighting their wider use.

The findings also underline the potential for additional gain in life expectancy through facilitating timely access to healthcare resources and improved prescribing practices. Poor access to healthcare resources contribute to increased mortality rates among people with SMI, [65] though UK estimates of reduced life expectancy in people with SMI are less dramatic than those arising from other healthcare systems[66] (possibly reflecting the universal access offered by the NHS). A confluence of clinician (e.g. insufficient assessment, poor communication, suboptimal prescribing habits), service providers (e.g. poor care coordination, insufficient funds), and patient (e.g. low motivation leading to poor adherence to treatment, difficulties in

understanding health care advice) factors may account for the poor access to and quality of health care among people with SMI. These factors all contribute to the reduced life expectancy among people with SMI, [67] as documented in this study. Recent research has confirmed disparities in access to specialist medical interventions (e.g. revascularisation) following a major health event among people with SMI.[68] Diagnostic delays related to physical comorbidities [69, 70] may also interfere with timely access to healthcare resources and effective treatment, increasing the risk of premature mortality. Our study findings suggest that tackling patient and health system barriers to accessing relevant healthcare resources and medical interventions would extend life expectancy by four to eight months across the life course within the broader SMI population.

Stigma and social isolation may influence mortality outcomes indirectly by impacting on lifestyle behaviours, treatment continuation or adherence, and the timely utilisation of healthcare resources.[71]. Stigma associated with SMI has been identified as a major barrier to accessing treatment and health care resources, and it occurs at all levels including healthcare professionals, society, and individuals.[72]. Similarly, socioeconomic inequalities in mortality outcomes among people with SMI are likely to reflect inequalities in access to healthcare resources, low health care-seeking behaviour, poorer knowledge of physical health symptoms and risk factors, and motivational challenges to accessing healthcare support. Our findings implied that the implementation of wider initiatives to tackle poverty and stigma would improve life expectancy among people with SMI by around 10%.

## Strengths and limitations of this study

To our knowledge this is the first study to consider both the population attribution fraction and the effectiveness of interventions in calculating the expected gain in life expectancy over the life course within specific SMI populations. Another major strength is the use of a multi-contextual approach to estimates potential gain in life expectancy over the lifecourse across specific SMI subgroups. Our estimation model shares limitations common to most projections of future gain in life expectancy, such as proportionality hypothesis (e.g. constant mortality rates and intervention effectiveness over time). We have partially addressed this concern by adopting short-term projections and conducting sensitivity analyses averaging mortality rates over a five-year period. Also, our estimates for treatment impact on life expectancy gains have been constrained to two drugs with the greatest evidence on harms and benefits within the SMI populations. Thus, our findings may not generalise to patients prescribed other antipsychotic or antidepressant drugs, or to polypharmacy. Given limited resources, it is not always possible to conduct studies on all issues in all settings prior to making a policy decision. This was also the case with the present study, where no single country collected relevant data on all study parameters estimated in the models. This concern places constraints about the generalizability of present study findings across different countries with varying healthcare and socioeconomic contexts. Ultimately, it is the responsibility of decision-makers to consider the extent to which evidence from one setting (UK) is transferrable to a different setting. Regardless, the framework set out in this study identified critical gaps in the current evidence base that may encourage future research and developments into modeling gains in life expectancy from addressing modifiable determinants of premature mortality among people with SMI. There were suggestions that PAF formula may overestimate excess burden of all-cause mortality due to smoking, [73] and this concern applies to our study estimates including the high collective estimates. Further, when confounders exist and one does not correct for this, the PAF is likely to be influenced. Moreover, residual confounding also exists and therefore the PAF suffers from overestimation. PAF estimates tend to vary, however, across populations, over time

and with different ranking of common mortality causes.[74] It also provides healthcare providers and policy makers with a useful tool to interpret the excess mortality due to specific factors among people with SMI.[74] The healthcare is a more complex system than conceptualised here, and our findings may underestimate the potential gain in life expectancy from healthcare-based interventions within the broader SMI population. Interventions at any level of the healthcare system, however, will be reflected in changes across other parts of the system including the healthcare characteristics considered in this study. Moreover, while the gap in life expectancy among men and women with SMI is similar (i.e. 13 and 12, respectively, years) future studies assessing possible differential gain in life expectancy among men and women with specific SMIs are warranted. When translating evidence from well-controlled trials into clinical practice, the dilution of the intervention effect is common. This concern is exacerbated by SMI patients representing a challenging group to treat, for several reasons (e.g. reduced insights into their condition, concerns around drugs side-effects). These concerns imply the need for greater efforts to accomplish the effects sizes used in our modeling approach. Several suggestions were put forward (e.g. improved mental and study physical care coordination, address patients' resistance to treatment or screening uptake) how this could be achieved, yet, evidence to confirm the feasibility of these proposals is currently lacking. Finally, our study estimates relied on UK-based data for mortality rates, while data on prevalence of modifiable risk factors and effectiveness of interventions coming mainly from developed countries. These estimates may not be directly transferrable to other countries with a different care system and epidemiology of risk factors social circumstances, or health care provision. We have aimed to moderate this concern by relying, whenever possible, on meta-analysis findings that accounts for variation in different study designs and methodology.

## Conclusion

Our study findings indicated that addressing unhealthy behaviours, suboptimal use of healthcare resources, and poor life circumstances have the potential prolong life expectancy at birth by four and six years within bipolar or schizophrenia disorders. If our study estimates are translatable into routine practice, we would expect around 24% (men) to 28% (women) narrowing in the life expectancy gap between the SMI and the general population from tackling modifiable risk factors. Given similar or greater life expectancy gaps in other Western countries these results suggest that at least comparable improvements in the longevity of people with SMI can be achieved elsewhere. Achieving these projections requires multisectoral approach to ensure that the complexity of SMI disorders are addressed at individual, clinical, and societal level.[75] The study findings need to be interpreted cautiously since translation of clinical trials evidence into routine care it is often challenging and ineffective.

## Supporting information

**S1 File.**
(DOCX)

## Author Contributions

**Conceptualization:** Alex Dregan, Ann McNeill, Fiona Gaughran, Peter B. Jones, Anna Bazley, Sean Cross, Kate Lillywhite, David Armstrong, Shubulade Smith, David P. J. Osborn, Robert Stewart, Til Wykes, Matthew Hotopf.

**Data curation:** Alex Dregan, David Armstrong, Robert Stewart, Matthew Hotopf.

**Formal analysis:** Alex Dregan, Peter B. Jones.

**Funding acquisition:** Matthew Hotopf.

**Investigation:** Alex Dregan, Ann McNeill, Fiona Gaughran, Peter B. Jones, Anna Bazley, Kate Lillywhite, David Armstrong, Shubulade Smith, David P. J. Osborn, Robert Stewart, Til Wykes, Matthew Hotopf.

**Methodology:** Alex Dregan, Ann McNeill, Fiona Gaughran, Peter B. Jones, Anna Bazley, Sean Cross, Kate Lillywhite, David Armstrong, Shubulade Smith, David P. J. Osborn, Robert Stewart, Til Wykes, Matthew Hotopf.

**Project administration:** Alex Dregan, Ann McNeill, Fiona Gaughran, Peter B. Jones, Kate Lillywhite, David Armstrong, Shubulade Smith, David P. J. Osborn, Robert Stewart, Til Wykes, Matthew Hotopf.

**Resources:** Robert Stewart, Til Wykes, Matthew Hotopf.

**Supervision:** Fiona Gaughran, Peter B. Jones, Anna Bazley, Sean Cross, Kate Lillywhite, David Armstrong, Shubulade Smith, David P. J. Osborn, Robert Stewart, Til Wykes, Matthew Hotopf.

**Validation:** Alex Dregan, Fiona Gaughran, Peter B. Jones, Anna Bazley, Sean Cross, Kate Lillywhite, David Armstrong, Shubulade Smith, David P. J. Osborn, Robert Stewart, Til Wykes, Matthew Hotopf.

**Visualization:** Alex Dregan, Ann McNeill, Fiona Gaughran, Peter B. Jones, Anna Bazley, Sean Cross, Kate Lillywhite, David Armstrong, Shubulade Smith, David P. J. Osborn, Robert Stewart, Til Wykes, Matthew Hotopf.

**Writing – original draft:** Alex Dregan.

**Writing – review & editing:** Ann McNeill, Fiona Gaughran, Peter B. Jones, Anna Bazley, Sean Cross, Kate Lillywhite, David Armstrong, Shubulade Smith, David P. J. Osborn, Robert Stewart, Til Wykes, Matthew Hotopf.

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
