## [Decision Letter · Decision Letter 0]

18 Nov 2019

PONE-D-19-27607

Potential gains in life expectancy from reducing amenable mortality among people diagnosed with serious mental illness

PLOS ONE

Dear Dr. Dregan,

Thank you for submitting your manuscript to PLOS ONE. After careful consideration, we feel that it has merit but does not fully meet PLOS ONE’s publication criteria as it currently stands. Therefore, we invite you to submit a revised version of the manuscript that addresses the points raised during the review process.

We would appreciate receiving your revised manuscript by Jan 02 2020 11:59PM. To enhance the reproducibility of your results, we recommend that if applicable you deposit your laboratory protocols in protocols.io, where a protocol can be assigned its own identifier (DOI) such that it can be cited independently in the future. For instructions see: http://journals.plos.org/plosone/s/submission-guidelines#loc-laboratory-protocols

We look forward to receiving your revised manuscript.

Kind regards,

Sinan Guloksuz, M.D., Ph.D.

Academic Editor

PLOS ONE

Journal Requirements:

Reviewers' comments:

Reviewer's Responses to Questions

**Comments to the Author**

1. Is the manuscript technically sound, and do the data support the conclusions?

Reviewer #1: No

Reviewer #2: Partly

2. Has the statistical analysis been performed appropriately and rigorously? 

Reviewer #1: No

Reviewer #2: Yes

3. Have the authors made all data underlying the findings in their manuscript fully available?

Reviewer #1: Yes

Reviewer #2: No

4. Is the manuscript presented in an intelligible fashion and written in standard English?

Reviewer #1: No

Reviewer #2: Yes

5. Review Comments to the Author

Reviewer #1: This is a study using population attributable risk factors to estimate potential gains in life expectancy in serious mental illnesses (schizophrenia, schizoaffective disorder and bipolar disorder).

1. This article is based on two assumptions: 1) the average effectiveness of a treatment intervention represent all studies and all countries, and 2) changes in mortality rate from adults with SMI in the United Kingdom applies to all countries.

2. I do not see the estimations of the effectiveness of the interventions for the various modifiable risks. Please add them to Table 1.

3. I am a clinician, so the idea that increasing the effectiveness of interventions related to lifestyle factors can increase life expectancy makes sense from a medical point of view. The only problem that I see is there is no acknowledgement that some of the lack of implementation of these interventions is due to the lack of cooperation of patients with SMI. These SMIs are associated with impairment in insight. You cannot force patients to do what they do not want to do. In summary, in an ideal system with an ideal physician or ideal health care, some of the lack of application of effective interventions may still be due to lack of interest or participation on the part of patients.

4. The idea of modifying healthcare factors and social factors is beyond what physicians do. These appear to be interventions at the political level. To mix healthcare interventions and politics does not seem wise to me. It appears to mix apples and oranges.

5. For a clinician like me, this article appears to be an exercise in mathematical modeling with limited applicability to the real world. Moreover, the title should reflect the fact that this mortality reduction applies only to the United Kingdom. It is laughable to think that these estimations have any value in countries with very different life expectancies or very different health systems. It also may not hurt to explain somehow in the title that this study was done using average rates of effectiveness from studies in multiple countries. Again, it is not likely that interventions such as reducing smoking or obesity in people with SMI would apply homogeneously across Western countries. Different Western countries are at different stages of change in the general population regarding these factors and the application of these interventions. If the data on interventions also includes other countries, such as those from Asia, that makes no sense at all. In summary, the data on interventions should come from the United Kingdom if you want to play to life expectancy in the United Kingdom. If the data on interventions comes from countries with very different life expectancies and different health systems, I do not see how this data can be used in a model based on the life expectancy of people with SMI in the United Kingdom.

6. The Discussion does not reflect awareness of the limitations of mathematical modeling.

7. “A recent trial illustrated, for instance, that a bespoke smoking cessation intervention embedded in routine mental health care settings (51) was associated with a 56% greater reduction in smoking rates compared to usual care within people with schizophrenia and bipolar disorder. This finding lends support to our proposed gain in life expectancy within the SMI population, if the effectiveness of current lifestyle interventions can be maintained or improved in the long-term.” This paragraph is a serious misrepresentation of that study and its follow-up study. Reference 51 is a pilot study that reports 12-month smoking cessation rates of 69% in 51 controls and 72% among 46 in the intervention group. Then there is a later study Gilbody S, Peckham E, Bailey D, Arundel C, Heron P, Crosland S, Fairhurst C, Hewitt C, Li J, Parrott S, Bradshaw T, Horspool M, Hughes E, Hughes T, Ker S, Leahy M, McCloud T, Osborn D, Reilly J, Steare T, Ballantyne E, Bidwell P, Bonner S, Brennan D, Callen T, Carey A, Colbeck C, Coton D, Donaldson E, Evans K, Herlihy H, Khan W, Nyathi L, Nyamadzawo E, Oldknow H, Phiri P, Rathod S, Rea J, Romain-Hooper CB, Smith K, Stribling A, Vickers C. Smoking cessation for people with severe mental illness (SCIMITAR+): a pragmatic randomised controlled trial. Lancet Psychiatry. 2019 May;6(5):379-390. doi: 10.1016/S2215-0366(19)30047-1.Epub 2019 Apr 8. PubMed PMID: 30975539; PubMed Central PMCID: PMC6546931. In this study, “The incidence of quitting at 6 months shows that smoking cessation can be achieved, but the waning of this effect by 12 months means more effort is needed for sustained quitting.” In summary, unfortunately, at 12 months the effect disappeared.

The most pessimistic interpretation is that we do not have any practical intervention for providing long-term smoking cessation in large groups of these patients. If you have any published intervention that has demonstrated that, please quote it. As indicated before, the pilot study quoted by the authors led to an unsuccessful trial. In my experience and through review of the long-term data in my state, some patients are able to stop on their own but, unfortunately, we as the health providers are not being very helpful.

8. Please delete the statement, “Our study findings corroborate with earlier evidence that effective mental healthcare would, in and of itself, be a potent means of reducing premature mortality by

addressing underlying symptoms and social problems arising from SMI.” This is not an independent study. You are making multiple assumptions using prior literature. It is not surprising that a mathematical model using prior literature supports the prior literature.

9. I think that clozapine and lithium are excellent drugs and should be used much more frequently. Many times, patients do not want to use them and you cannot force them to take them. They are generic drugs that are not promoted by pharmaceutical companies. Moreover, they are mainly started and mainly managed by psychiatrists due to their complex pharmacology. My psychiatry residents do not know how to prescribe them since most of the attendings in my academic department do not use them. Thus, I am not optimistic that in the future they will be prescribed more frequently, at least not in my state in the US.

10. There are many studies on the barriers involved in the use of clozapine. “Verdoux H, Quiles C, Bachmann CJ, Siskind D. Prescriber and institutional barriers and facilitators of clozapine use: A systematic review. Schizophr Res. 2018 Nov;201:10-19. doi: 10.1016/j.schres.2018.05.046. Epub 2018 Jun 4. PubMed PMID: 29880453.” The truth is that people like me, who consider themselves experts on clozapine, appear to be incompetent in overcoming these barriers where they practice. It would be helpful if the authors would teach us how to increase the use of clozapine or lithium. They appear to know things that we do not know.

11. The Limitations do not reflect any of the prior limitations of using data on the effectiveness of intervention from many countries and then applying it to the life expectancy of people with SMI in the United Kingdom and then trying to generalize it to the whole world.

12. There is no attempt to consider the lack of cooperation of patients and physicians in improving the dismal situation surrounding the life expectancy of people with SMI. I work as a consultant in the public system of a state in the US. The first problem for me in implementing basic interventions such as increasing the use of clozapine and lithium is that some clinicians do not want to deal with their complications and, in the case of clozapine, with much more paperwork. Once I am dealing with convinced and trained clinicians, they need to convince each individual patient and their families. Nobody is paying for advertisements for these two generic drugs. Pharmaceutical companies support other antipsychotics and other mood stabilizers that are competing with clozapine and lithium. I would like to live in the same mathematical universe as the authors and believe that in my state these two drugs will be more widely prescribed because it is the right thing to do.

13. Please understand that I do not deny that the authors have very good intentions, but estimating the effect of interventions without considering the barriers does not appear very useful in the real world. On the other hand, I acknowledge that modeling the barriers to implementation will not be easy.

Reviewer #2: This is an important manuscript to enhance implementation of effective treatments for modifiable risk factors. The authors describe an important effort to summarize literature and calculate with the numbers from previous studies.

However, I have some points to consider:

Abstract:

- please mention the timeframe of the data that was used to calculate your results.

-conclusions: These % are under ideal circumstances and without the limitation of overestimation which often comes with PAFs, please rephrase this cautiously.

Introduction:

-these diseases are party attributable,. Obesity can also be caused by olanzapine and clozapine and therefore these factors might not be as easily tackled as we might wish.

Methods:

-Why not update the literature beyond 2018? For example, a recent study found an increasing number of years life lost https://www.ncbi.nlm.nih.gov/pubmed/30446270

Also, the results of the scimitar trial regarding smoking are recently published which gives important nuances in how hard treatment is in these groups.

-Again, modifiable risk factors: treatment with certain antipsychotics induces the risk of cardiovasculair disease (see De Hert 2012, nature reviews) and therefore for example is less modifiable than we hope. This should be at least mentioned if one cannot correct for this in some way in the analyses.

-The use of PAFs and formula has several limitations and overestimation is likely to occur

https://www.ncbi.nlm.nih.gov/pmc/articles/PMC4339639/

Please elaborate on the choice for the PAF and why this particular formal was chosen, here and/or in the limitations section of the manuscript.

- 36% for smoking cessation is likely to be an overestimation (see scimitair results british journal of psychiatry Gilbody et al.)

Results:

Is it possible to correct for the interaction of the factors in Table 1 and the RRs for mortality?

Healthcare system determinants: What about the increased risk for adverse effects that come with lithium and antipsychotics, does this balance out against the gains?

Collective estimates: 90% seems high, is there a possibility of overestimation?

Discussion

Indeed, standard interventions are less effective, more effort Is needed to accomplish similar effect sizes. SMI patients are a harder to treat population, please elaborate on this and how we can improve our interventions.

Please add the long-term meta-analysis on antipsychotics/clozapine and mortality to the litertarue (ref56-58)

Limitations: When confounders exist and one does not correct for this, the PAF is likely to be influenced. Moreover, residual confounding also exists and therefore the PAF suffers from overestimation. Please mention this here.

What are the limitations regarding generalizability to non-western countries?

6. PLOS authors have the option to publish the peer review history of their article (what does this mean?). If published, this will include your full peer review and any attached files.

Reviewer #1: No

Reviewer #2: Yes: Jentien Vermeulen, MD PhD

---

## [Author Response · Author response to Decision Letter 0]

12 Feb 2020

REVIEWER #1 COMMENTS: 

1. This article is based on two assumptions: 1) the average effectiveness of a treatment intervention represent all studies and all countries, and 2) changes in mortality rate from adults with SMI in the United Kingdom applies to all countries.

R: The Reviewer makes a valid observation about our study methodological assumption. We have added the following sentences in the Discussion section on Page 16: “The present study modeling approach relied on two assumptions that (a) the estimated average effectiveness of interventions represented all studies and countries, and (b) that mortality rates of UK adults with SMI applied universally.” 

2. I do not see the estimations of the effectiveness of the interventions for the various modifiable risks. Please add them to Table 1.

R: Please note that the estimations of the effectiveness of the interventions for the modifiable risks are included in Table 2, second column (ES).

3. I am a clinician, so the idea that increasing the effectiveness of interventions related to lifestyle factors can increase life expectancy makes sense from a medical point of view. The only problem that I see is there is no acknowledgement that some of the lack of implementation of these interventions is due to the lack of cooperation of patients with SMI. These SMIs are associated with impairment in insight. You cannot force patients to do what they do not want to do. In summary, in an ideal system with an ideal physician or ideal health care, some of the lack of application of effective interventions may still be due to lack of interest or participation on the part of patients.

R: We are now stating in the Discussion section on Pag 15, that: “The implementation of lifestyle interventions depends also on achieving increased acceptance and participation rates on the part of patients. SMIs are commonly associated with impairment in patients’ awareness about the implications of unhealthy behaviours or treatment non-adherence to their health and wellbeing. Poor insight into the illness and its treatment, may challenge clinicians’ capacity to convince SMI patients to adopt healthy behaviours or adhere to treatment. Future studies are needed to identify how best to enhance SMI patients’ awareness about the benefits of treatment compliance and lifestyle changes to their health and quality of life.” 

Please also note that on Page 18 we did mention that several factors contribute to challenges in implementing lifestyle-oriented interventions: “A confluence of clinician (e.g. insufficient assessment, poor communication, suboptimal prescribing habits), service providers (e.g. poor care coordination, insufficient funds), and patient (e.g. low motivation leading to poor adherence to treatment, difficulties in understanding health care advice) factors may account for the poor access to and quality of health care among people with SMI.”

4. The idea of modifying healthcare factors and social factors is beyond what physicians do. These appear to be interventions at the political level. To mix healthcare interventions and politics does not seem wise to me. It appears to mix apples and oranges.

 R: While we agree that social factors are beyond physicians’ usual responsibilities, we respectfully disagree with the Reviewer suggestion that mixing healthcare and politics are necessarily opposites. To bring about sustainable health and life-expectancy benefits to patients with SMI requires multisystem interventions involving multiple stakeholders, including patients, clinicians, health care systems, policy makers, researchers, and employers. In this respect, we did state throughout the Discussion section that multiple stakeholders need to be involved in improving life expectancy in people with SMI, beyond physicians. For instance, on Page 18 we have stated that: “Our study findings suggest that tackling patient and health system barriers to accessing relevant healthcare resources and medical interventions would extend life expectancy by four to eight months across the life course within the broader SMI population.” On Page 19 we have acknowledged that: “Our findings implied that the implementation of wider initiatives to tackle poverty and stigma would improve life expectancy among people with SMI by around 10%.” Further, the last sentence of the Conclusion stated: “Achieving these projections requires multisectoral approach to ensure that the complexity of SMI disorders are addressed at individual, clinical, and societal level.”

5. For a clinician like me, this article appears to be an exercise in mathematical modeling with limited applicability to the real world. Moreover, the title should reflect the fact that this mortality reduction applies only to the United Kingdom. It is laughable to think that these estimations have any value in countries with very different life expectancies or very different health systems. It also may not hurt to explain somehow in the title that this study was done using average rates of effectiveness from studies in multiple countries. Again, it is not likely that interventions such as reducing smoking or obesity in people with SMI would apply homogeneously across Western countries. Different Western countries are at different stages of change in the general population regarding these factors and the application of these interventions. If the data on interventions also includes other countries, such as those from Asia, that makes no sense at all. In summary, the data on interventions should come from the United Kingdom if you want to play to life expectancy in the United Kingdom. If the data on interventions comes from countries with very different life expectancies and different health systems, I do not see how this data can be used in a model based on the life expectancy of people with SMI in the United Kingdom.

R: The Reviewer makes a valid point about the challenges associated with applicability and transferability of findings across different healthcare systems. To address this concern we have relied, whenever possible, on data from meta-analyses. We have included the following statement on Page 19: “Given limited resources, it is not always possible to conduct studies on all issues in all settings prior to making a policy decision. This was also the case with the present study, where no single country collected relevant data on all study parameters estimated in the models. This concern places constraints about the generalizability of present study findings across different countries with varying healthcare and socioeconomic contexts. Ultimately, it is the responsibility of decision-makers to consider the extent to which evidence from one setting (UK) is transferrable to a different setting. Regardless, the framework set out in this study identified critical gaps in the current evidence base that may encourage future research and developments into modeling gains in life expectancy from addressing modifiable determinants of premature mortality among people with SMI.” We have also changed the study title to: “Potential gains in life expectancy from reducing amenable mortality among people diagnosed with serious mental illness in the United Kingdom.” 

6. The Discussion does not reflect awareness of the limitations of mathematical modeling.

R: Please see our reply to Reviewer 2, Points, 5, 7, and 13 which discuss the concerns around PAF estimates reliability and the lack of modeling for interaction effects. 

7. “A recent trial illustrated, for instance, that a bespoke smoking cessation intervention embedded in routine mental health care settings (51) was associated with a 56% greater reduction in smoking rates compared to usual care within people with schizophrenia and bipolar disorder. This finding lends support to our proposed gain in life expectancy within the SMI population, if the effectiveness of current lifestyle interventions can be maintained or improved in the long-term.” This paragraph is a serious misrepresentation of that study and its follow-up study. Reference 51 is a pilot study that reports 12-month smoking cessation rates of 69% in 51 controls and 72% among 46 in the intervention group. Then there is a later study Gilbody S, Peckham E, Bailey D, Arundel C, Heron P, Crosland S, Fairhurst C, Hewitt C, Li J, Parrott S, Bradshaw T, Horspool M, Hughes E, Hughes T, Ker S, Leahy M, McCloud T, Osborn D, Reilly J, Steare T, Ballantyne E, Bidwell P, Bonner S, Brennan D, Callen T, Carey A, Colbeck C, Coton D, Donaldson E, Evans K, Herlihy H, Khan W, Nyathi L, Nyamadzawo E, Oldknow H, Phiri P, Rathod S, Rea J, Romain-Hooper CB, Smith K, Stribling A, Vickers C. Smoking cessation for people with severe mental illness (SCIMITAR+): a pragmatic randomised controlled trial. Lancet Psychiatry. 2019 May;6(5):379-390. doi: 10.1016/S2215-0366(19)30047-1.Epub 2019 Apr 8. PubMed PMID: 30975539; PubMed Central PMCID: PMC6546931. In this study, “The incidence of quitting at 6 months shows that smoking cessation can be achieved, but the waning of this effect by 12 months means more effort is needed for sustained quitting.” In summary, unfortunately, at 12 months the effect disappeared.

The most pessimistic interpretation is that we do not have any practical intervention for providing long-term smoking cessation in large groups of these patients. If you have any published intervention that has demonstrated that, please quote it. As indicated before, the pilot study quoted by the authors led to an unsuccessful trial. In my experience and through review of the long-term data in my state, some patients are able to stop on their own but, unfortunately, we as the health providers are not being very helpful.

R: We thank you the reviewer for this valuable insight. We have responded on a similar concern raised by Reviewer 2, Point 6 related to the Gilbody et al. Scimitar trial which has been added as a reference in the study. 

8. Please delete the statement, “Our study findings corroborate with earlier evidence that effective mental healthcare would, in and of itself, be a potent means of reducing premature mortality by

addressing underlying symptoms and social problems arising from SMI.” This is not an independent study. You are making multiple assumptions using prior literature. It is not surprising that a mathematical model using prior literature supports the prior literature.

R: As per Reviewer comment, we have now excluded the statement from the manuscript. 

9. I think that clozapine and lithium are excellent drugs and should be used much more frequently. Many times, patients do not want to use them and you cannot force them to take them. They are generic drugs that are not promoted by pharmaceutical companies. Moreover, they are mainly started and mainly managed by psychiatrists due to their complex pharmacology. My psychiatry residents do not know how to prescribe them since most of the attendings in my academic department do not use them. Thus, I am not optimistic that in the future they will be prescribed more frequently, at least not in my state in the US.

R: We share Reviewer’ opinion on clozapine and lithium, however these concerns are not unique to these drugs and innovation in medicine would scarcely progress if we adopted such a pessimistic stance. We have revised Page 18 to state that: “While effective, clozapine and lithium are not always accepted by patients and require expertise and experience in order to be prescribed safely. Thus, there are implementation challenges to any policy highlighting their wider use.” 

10. There are many studies on the barriers involved in the use of clozapine. “Verdoux H, Quiles C, Bachmann CJ, Siskind D. Prescriber and institutional barriers and facilitators of clozapine use: A systematic review. Schizophr Res. 2018 Nov;201:10-19. doi: 10.1016/j.schres.2018.05.046. Epub 2018 Jun 4. PubMed PMID: 29880453.” The truth is that people like me, who consider themselves experts on clozapine, appear to be incompetent in overcoming these barriers where they practice. It would be helpful if the authors would teach us how to increase the use of clozapine or lithium. They appear to know things that we do not know.

 R: We have emphasized throughout the manuscript the about the need for multisectorial and multifaceted efforts to address concerns around the gap in life expectancy among people with SMI. For instance, on Page 18 we have stated: “A confluence of clinician (e.g. insufficient assessment, poor communication, suboptimal prescribing habits), service providers (e.g. poor care coordination, insufficient funds), and patient (e.g. low motivation leading to poor adherence to treatment, difficulties in understanding health care advice) factors may account for the poor access to and quality of health care among people with SMI. These factors all contribute to the reduced life expectancy among people with SMI, (64) as documented in this study.” On Page 21 we have stated: “Our study findings indicated that addressing unhealthy behaviours, suboptimal use of healthcare resources, and poor life circumstances have the potential prolong life expectancy at birth by four and six years within bipolar or schizophrenia disorders.” 

11. The Limitations do not reflect any of the prior limitations of using data on the effectiveness of intervention from many countries and then applying it to the life expectancy of people with SMI in the United Kingdom and then trying to generalize it to the whole world.

R: We have included the following sentences on Page 21 of the Limitations sections: “The present study estimates and findings relied on UK-related data for mortality rates, while data on prevalence of modifiable risk factors and effectiveness of interventions coming mainly from developed countries. These estimates may not be directly transferrable to other countries with a different care system and epidemiology of risk factors social circumstances, or health care provision. We have aimed to moderate this concern by relying, whenever possible, on meta-analysis findings that accounts for variation in different study designs and methodology.” 

12. There is no attempt to consider the lack of cooperation of patients and physicians in improving the dismal situation surrounding the life expectancy of people with SMI. I work as a consultant in the public system of a state in the US. The first problem for me in implementing basic interventions such as increasing the use of clozapine and lithium is that some clinicians do not want to deal with their complications and, in the case of clozapine, with much more paperwork. Once I am dealing with convinced and trained clinicians, they need to convince each individual patient and their families. Nobody is paying for advertisements for these two generic drugs. Pharmaceutical companies support other antipsychotics and other mood stabilizers that are competing with clozapine and lithium. I would like to live in the same mathematical universe as the authors and believe that in my state these two drugs will be more widely prescribed because it is the right thing to do.

R: The Reviewer may have misunderstood the purpose of our paper. We are testing out areas where there might be tractable targets for intervention to address the health disparity (increased mortality in SMI). We do not pretend that the interventions proposed can easily be implemented, but we do think that it is reasonable to model the potential gains where one might have the greatest traction. We share the Reviewer’s concerns around dealing with complex patients and challenges around non-adherence with treatment/control recommendations, which we consider are highly context specific.

 13. Please understand that I do not deny that the authors have very good intentions, but estimating the effect of interventions without considering the barriers does not appear very useful in the real world. On the other hand, I acknowledge that modeling the barriers to implementation will not be easy.

R: Thank you, this point is addressed above under point 12. We naturally understand the concern of the Reviewer and have added the following sentence in the Conclusion section, Page 21: “The study findings need to be interpreted cautiously since translation of clinical trials evidence into routine care it is often challenging and ineffective.” 

Reviewer #2 (Comments to Author): 

Summary: 

This is an important manuscript to enhance implementation of effective treatments for modifiable risk factors. The authors describe an important effort to summarize literature and calculate with the numbers from previous studies.

R: Thank you for the positive evaluation of our research.

Specific Comments: 

1. Abstract:

- please mention the timeframe of the data that was used to calculate your results.

-conclusions: These % are under ideal circumstances and without the limitation of overestimation which often comes with PAFs, please rephrase this cautiously.

R: We are now clarifying that the timeframe of the data as: “The predicted estimates were based on mortality rates for year 2014-2015.“ We have now included the following text in the Abstract: “These projections represent ideal circumstances and without the limitation of overestimation which often comes with PAFs.“

2. Introduction:

-these diseases are party attributable,. Obesity can also be caused by olanzapine and clozapine and therefore these factors might not be as easily tackled as we might wish.

R: Clarified on Page 3, Introduction that these diseases were partly attributable. 

3. Methods:

-Why not update the literature beyond 2018? For example, a recent study found an increasing number of years life lost https://www.ncbi.nlm.nih.gov/pubmed/30446270. Also, the results of the scimitar trial regarding smoking are recently published which gives important nuances in how hard treatment is in these groups.

R: We thank the Reviewer for the helpful information. We have added the following text on Page 18: “These suggestions are supported by recent evidence documenting that people with SMI failed to manifest similar reduction in mortality rates due to natural causes observed in the general populations over the past decades.(46)”

4. -Again, modifiable risk factors: treatment with certain antipsychotics induces the risk of cardiovasculair disease (see De Hert 2012, nature reviews) and therefore for example is less modifiable than we hope. This should be at least mentioned if one cannot correct for this in some way in the analyses.

R: Please note that on Page 18 we did mention that antipsychotics have adverse effects on cardiometabolic disorders. We have now added the De Hert et al. (2012) reference on Page 18:” Despite valid concerns(53-57) about a potential association between antipsychotic drugs with increased risk of metabolic disorders, …”. We have also added the De Hert et al. reference on Page 19 where have stated that:” This may seem counter-intuitive given that clozapine is prone to cause obesity, diabetes, cardiomyopathy(57) and…”.

5. -The use of PAFs and formula has several limitations and overestimation is likely to occur

https://www.ncbi.nlm.nih.gov/pmc/articles/PMC4339639/ Please elaborate on the choice for the PAF and why this particular formal was chosen, here and/or in the limitations section of the manuscript.

R: We are now clarifying on Page 21 that: “There are suggestions that PAF formula may overestimate excess burden of all-cause mortality due to smoking, (72) and this concern applies to our study estimates as well. PAF estimates tend to vary, however, across populations, over time and with different ranking of common mortality causes.(73) It also provides healthcare providers and policy makers with a useful tool to interpret the excess mortality due to specific factors among people with SMI.(73)” 

6. - 36% for smoking cessation is likely to be an overestimation (see scimitair results british journal of psychiatry Gilbody et al.)

R: We have now clarified on Page 17 that: “Other studies suggested, however, lower rates of smoking cessation.(53) Regardless of these variation, such…”

7. Results:

Is it possible to correct for the interaction of the factors in Table 1 and the RRs for mortality?

R: Please note that on Page 6 we did state: “To allow for the overlap between different modifiable risk factors in influencing all-cause mortality, we have also estimated the combined impact of multiple modifiable risk factors within each group of determinants (e.g. behavioural, healthcare, social) and across all modifiable risk factors… While the formula assumes no interaction effects, it ensures that the PAF for the combined contribution of modifiable risk factors to all-cause mortality is not greater than 1.(28) “ 

8. Healthcare system determinants: What about the increased risk for adverse effects that come with lithium and antipsychotics, does this balance out against the gains?

R: We agree that this a valid point, yet very challenging to estimate in practice. We are not aware of any multi-component trial that has considered the simultaneous effects of healthcare and therapeutic pathways on excess mortality within SMI. We do not think that the evidence suggests that adverse effects of lithium and clozapine balance out the gains – otherwise we would not be suggesting these as potential interventions. As stated on Page 17: “Despite valid concerns(54-58) about a potential association between antipsychotic drugs with increased risk of metabolic disorders, previous studies revealed a potential for clozapine and lithium to reduce premature mortality among people with schizophrenia and bipolar disorders.(59-61).”

9. Collective estimates: 90% seems high, is there a possibility of overestimation?

R: Please note that these estimates are context specific and our study has considered most common modifiable factors supported by the existing literature. We have included the following clarification on Page 20: “…including the high (90%) collective estimates.” 

10. Discussion

Indeed, standard interventions are less effective, more effort Is needed to accomplish similar effect sizes. SMI patients are a harder to treat population, please elaborate on this and how we can improve our interventions.

R: We are now saying on Page 21 that: “When translating evidence from well-controlled trials into clinical practice, the dilution of the intervention effect is common. This concern is exacerbated by SMI patients representing a challenging group to treat, for several reasons (e.g. reduced insights into their condition, concerns around drugs side-effects). These concerns imply the need for greater efforts to accomplish the effects sizes used in our modeling approach. Several suggestions were put forward (e.g. improved mental and study physical care coordination, address patients’ resistance to treatment or screening uptake) how this could be achieved, yet, evidence to confirm the feasibility of these proposals is currently lacking.” 

11. Please add the long-term meta-analysis on antipsychotics/clozapine and mortality to the litertarue (ref56-58)

R: As per Reviewer suggestion we have now added De Hert et al. meta-analysis study on antipsychotics/clozapine. 

 12. Limitations: When confounders exist and one does not correct for this, the PAF is likely to be influenced. Moreover, residual confounding also exists and therefore the PAF suffers from overestimation. Please mention this here.

R: We have added the following sentence on Page 20: “Further, when confounders exist and one does not correct for this, the PAF is likely to be influenced. Moreover, residual confounding also exists and therefore the PAF suffers from overestimation.”

13. What are the limitations regarding generalizability to non-western countries?

R. Please see our reply to Reviewer 1, Point 11. 

EDITORIAL REQUIREMENTS 

R: Our revised ms meets the PLOS ONE’s style requirements. 

Please subject Figure 1 at a minimum resolution of >350 ppi or in one of the following VECTOR formats: .ai, .docx, .emf, .eps, .pdf 

R: We have adddressed the requirements for the Supplementary material and have enhanced the resolution of Figure 1.

Statistical Graphs 

For Figure 2, please provide a graph output directly from the software used to create it in an editable VECTOR file format, such as .wmf or .eps, or as an Excel graph, if created in Excel. If you provide .pdf files, be sure that these are in VECTOR file formal, not Raster file format. Raster (picture) files, such as .jpg or .tif, output directly or embedded in vector files, are not acceptable. All statistical graphs in accepted manuscripts are recreated in-house. 

R: We have saved and attached Figure 2 as an .wmf file

References:

1. Ridker PM, Buring JE, Cook NR, et al. C-reactive protein, the metabolic syndrome, and risk of incident cardiovascular events: an 8-year follow-up of 14 719 initially healthy American women. Circulation 2003;107(3):391-7. doi: 10.1161/01.cir.0000055014.62083.05 [published Online First: 2003/01/29]

2. Sesso HD, Buring JE, Rifai N, et al. C-reactive protein and the risk of developing hypertension. JAMA 2003;290(22):2945-51. doi: 10.1001/jama.290.22.2945 [published Online First: 2003/12/11]

3. Groenwold RH, Sterne JA, Lawlor DA, et al. Sensitivity analysis for the effects of multiple unmeasured confounders. Ann Epidemiol 2016;26(9):605-11. doi: 10.1016/j.annepidem.2016.07.009 [published Online First: 2016/09/01]

4. Yusuf S, Hawken S, Ounpuu S, et al. Effect of potentially modifiable risk factors associated with myocardial infarction in 52 countries (the INTERHEART study): case-control study. Lancet 2004;364(9438):937-52. doi: 10.1016/S0140-6736(04)17018-9 [published Online First: 2004/09/15]

5. Hare DL, Toukhsati SR, Johansson P, et al. Depression and cardiovascular disease: a clinical review. Eur Heart J 2014;35(21):1365-72. doi: 10.1093/eurheartj/eht462 [published Online First: 2013/11/28]

6. Khambaty T, Stewart JC, Gupta SK, et al. Association Between Depressive Disorders and Incident Acute Myocardial Infarction in Human Immunodeficiency Virus-Infected Adults: Veterans Aging Cohort Study. JAMA Cardiol 2016;1(8):929-37. doi: 10.1001/jamacardio.2016.2716 [published Online First: 2016/08/25]

7. Whooley MA. Depression and cardiovascular disease: healing the broken-hearted. JAMA 2006;295(24):2874-81. doi: 10.1001/jama.295.24.2874 [published Online First: 2006/06/29]

8. Alberti KG, Eckel RH, Grundy SM, et al. Harmonizing the metabolic syndrome: a joint interim statement of the International Diabetes Federation Task Force on Epidemiology and Prevention; National Heart, Lung, and Blood Institute; American Heart Association; World Heart Federation; International Atherosclerosis Society; and International Association for the Study of Obesity. Circulation 2009;120(16):1640-5. doi: 10.1161/CIRCULATIONAHA.109.192644 [published Online First: 2009/10/07]

9. Millasseau SC KR, Ritter JM, Chowienczyk PJ. Determination of age‐related increases in large artery stiffness by digital pulse contour analysis. Clinical science (London) 2002;103:371-77.

10. Lewis TT, Sutton-Tyrrell K, Penninx BW, et al. Race, psychosocial factors, and aortic pulse wave velocity: the Health, Aging, and Body Composition Study. J Gerontol A Biol Sci Med Sci 2010;65(10):1079-85. doi: 10.1093/gerona/glq089 [published Online First: 2010/06/05]

11. Tiemeier H, Breteler MM, van Popele NM, et al. Late-life depression is associated with arterial stiffness: a population-based study. J Am Geriatr Soc 2003;51(8):1105-10. [published Online First: 2003/08/02]

12. Seldenrijk A, Vogelzangs N, van Hout HP, et al. Depressive and anxiety disorders and risk of subclinical atherosclerosis Findings from the Netherlands Study of Depression and Anxiety (NESDA). J Psychosom Res 2010;69(2):203-10. doi: 10.1016/j.jpsychores.2010.01.005 [published Online First: 2010/07/14]

13. Camacho A, McClelland RL, Delaney JA, et al. Antidepressant Use and Subclinical Measures of Atherosclerosis: The Multi-Ethnic Study of Atherosclerosis. J Clin Psychopharmacol 2016;36(4):340-6. doi: 10.1097/JCP.0000000000000518 [published Online First: 2016/06/09]

14. Valeri L, Vanderweele TJ. Mediation analysis allowing for exposure-mediator interactions and causal interpretation: theoretical assumptions and implementation with SAS and SPSS macros. Psychol Methods 2013;18(2):137-50. doi: 10.1037/a0031034 [published Online First: 2013/02/06]

---

## [Editor Report · Decision Letter 1]

6 Mar 2020

Potential gains in life expectancy from reducing amenable mortality among people diagnosed with serious mental illness in the United Kingdom

PONE-D-19-27607R1

Dear Dr. Dregan,

We are pleased to inform you that your manuscript has been judged scientifically suitable for publication and will be formally accepted for publication once it complies with all outstanding technical requirements.

With kind regards,

Sinan Guloksuz, M.D., Ph.D.

Academic Editor

PLOS ONE
---

## [Editor Report · Acceptance letter]

11 Mar 2020

PONE-D-19-27607R1 

Potential gains in life expectancy from reducing amenable mortality among people diagnosed with serious mental illness in the United Kingdom 

Dear Dr. Dregan:

I am pleased to inform you that your manuscript has been deemed suitable for publication in PLOS ONE. Congratulations! Your manuscript is now with our production department. 

With kind regards,

on behalf of

Dr. Sinan Guloksuz 

Academic Editor

PLOS ONE